

# Experimental evaluation of genomic DNA degradation rates for the pathogen *Pseudogymnoascus destructans* (Pd) in bat guano

Jenny Urbina[1], Tara Chestnut[2], Donelle Schwalm[3], Jenn Allen[1] and Taal Levi[1]

[1] Department of Fisheries and Wildlife, Oregon State University, Corvallis, OR, United States of America
[2] Mount Rainier National Park, National Park Service, Ashford, WA, United States of America
[3] Department of Biology, University of Maine-Farmington, Farmington, ME, United States of America

## ABSTRACT

*Pseudogymnoascus destructans* (Pd), the causative agent of white-nose syndrome in bats (WNS), has led to dramatic declines of bat populations in eastern North America. In the spring of 2016, WNS was first detected at several locations in Washington State, USA, which has prompted the need for large scale surveillance efforts to monitor the spread of Pd. Pd is typically detected in bats using invasive methods requiring capturing and swabbing individual bats. However, Pd can also be detected in guano, which may provide an efficient, affordable, and noninvasive means to monitor Pd in bats across North America. The widespread implementation of Pd surveillance in guano is hindered by substantial uncertainty about the probability of detecting Pd when present, and how this probability is influenced by the time since defecation, local environmental conditions, the amount of guano sampled, and the original concentration of DNA shed in the guano. In addition, the expected degradation rate of Pd DNA depends on whether the Pd DNA found in guano represents extracellular DNA fragments, intracellular DNA from dead Pd fungal cells, or from intracellular and viable Pd cells. While this is currently unknown, it has been posited that most environmental DNA, such as Pd found in guano long after defecation, is fragmented extracellular DNA. Using non-viable isolated DNA at precise quantities, we experimentally characterized the degradation rates of Pd DNA in guano samples. We spiked 450 guano samples with Pd gDNA in a 10-fold dilution series from 1 million to 1,000 fg and placed them in variable environmental conditions at five sites at Mount Rainier National Park in Washington State, which is a priority location for Pd surveillance. We evaluated DNA degradation over 70 days by quantifying the amount of DNA in samples collected every 14 days using real-time quantitative PCR (qPCR). Our sampling period was from July 10th to September 17th 2018 which overlaps with bat movement between summer roosts as well as movement from maternity colonies fall swarms. We detected Pd DNA in guano 56 and 70 days after inoculation with 1 million and 100,000 fg respectively, while the lowest quantity (1,000 fg) was detected until 42 days. Detection probability was variable among sites and lower where samples were left exposed without overhead cover. If Pd is shed as extracellular DNA in guano at quantities above 1,000 fg, then guano collection is likely to provide an effective tool for environmental screening of Pd that can be employed in an early detection and rapid response framework throughout Washington and other regions where this disease is rapidly emerging.

Corresponding author
Jenny Urbina,
jenny.gonzalez@oregonstate.edu

## INTRODUCTION

Emerging infectious diseases (EIDs) are one of many drivers of the current biodiversity crisis (*Harvell et al., 1999*; *Daszak, Cunningham & Hyatt, 2003*; *Tompkins et al., 2015*; *Cunningham, Daszak & Wood, 2017*; *Reid et al., 2019*). In eastern North America, white-nose syndrome (WNS) is among the most dramatic of recent emergence events. WNS, which is caused by the psychrophilic fungus *Pseudogymnoascus destructans* (Pd), has led to unprecedented mortality of hibernating bats (*Reeder & Moore, 2013*) and now threatens several species with regional extinction (*Frick et al., 2010*; *Frick et al., 2017*). The emergence of Pd in North America was first reported from New York in 2006 and has since spread to 38 states and seven Canadian provinces (http://www.whitenosesyndrome.org, accessed September 15, 2019). Pd has been confirmed in 20 bat species from the United States and Canada, including two endangered and one threatened species (http://www.whitenosesyndrome.org, accessed September 15, 2019). The first case of WNS in western North America was detected in 2016 (*Lorch et al., 2016*) in King County, Washington from a little brown bat (*Myotis lucifugus*) and a silver-haired bat (*Lasionycteris noctivagans*) tested positive for the fungus but had no clinical signs of the disease (https://wdfw.wa.gov/species-habitats/diseases/bat-white-nose#, accessed September 15, 2019). In April 2017, Pd was confirmed from a Yuma myotis bat (*Myotis yumanensis*) in King County, followed by Pd detections from two little brown bats and two Yuma myotis in Lewis County, Washington in May 2017 with no indication of disease. In 2018, Pd and WNS were confirmed in little brown bats and Yuma myotis from several additional sites in King County. Recently, in March 2019, WNS was confirmed positive for the first time in North America in a western long-eared myotis (*Myotis evotis*), and the first case of WNS outside of King County was confirmed from a little brown bat in Pierce County.

Efforts to document the spread and impacts on WNS typically employ methods that include capture and handling of individual bats to collect skin swabs to test for Pd presence, and collecting skin biopsies when clinical signs of WNS are observed. Alternatively, noninvasive detection from guano can be used to detect the presence of pathogens (*Kriger, Hero & Ashton, 2006*; *Oehm et al., 2011*) like Pd (*Langwig et al., 2016*; *Dobony & Johnson, 2018*). Pd in guano can be isolated and quantified with methods such as real-time qPCR or droplet digital PCR (*Lorch et al., 2013*). Guano samples can additionally be used to identify the species of bat host, which can be difficult to determine by morphology alone (*Höss et al., 1992*; *Kurose, Masuda & Tatara, 2005*; *Player et al., 2017*), as well as bat diet (*Walker et al., 2016*).

The widespread implementation of Pd surveillance in guano is hindered by substantial uncertainty about the probability of detecting Pd when present, and how this probability is influenced by the time since defecation, local environmental conditions, the amount of guano sampled, and the original concentration of DNA shed in the guano. In addition,

the expected degradation rate of Pd DNA depends on whether the Pd DNA found in guano represents extracellular DNA fragments, intracellular DNA from dead Pd fungal cells, or from intracellular and viable Pd cells. While this is currently unknown, it is posited that most environmental DNA, such as Pd found in guano long after defecation (*Brownlee-Bouboulis & Reeder, 2013*), is fragmented extracellular DNA, which can persist for long periods when bound to soil colloids and partially protected from degradation (*Agnelli et al., 2007*).

To inform interpretation of noninvasive surveillance of Pd using guano, we used experimental guano inoculation in natural field settings to assess the probability of detecting Pd DNA and the rate of DNA degradation. The objective of this experiment was to understand how long we can detect extracellular Pd DNA in bat guano and how environmental factors affect detection. We evaluated DNA degradation over 70 days by quantifying the amount of DNA samples collected every 14 days. We hypothesized probability of detection of Pd would decrease over time, and would vary based on the mass of collected sample, and exposure to ambient conditions. The results of our study will inform the development of future protocols and monitoring actions for bats and Pd/WNS spread.

## MATERIALS & METHODS

### Experimental setup

Five different locations in Mount Rainier National Park were selected as representative regions where active bat monitoring is occurring (Fig. 1). All locations were buildings and structures that were occupied by bats. Two sites were in the Nisqually River watershed, two in the Ohanapecosh River watershed and one was in the Carbon River watershed. Sites were categorized as exposed or protected according to the presence of structures shielding guano samples from direct solar radiation or rain. Two sites, in the Carbon and Nisqually watersheds, were protected with an overhead cover to avoid sun exposure and rain, while the remaining sites were completely exposed (Fig. 1). The regional weather in the study area during our study presented the following conditions: Temperature average low = 5.6 C and average high = 16 C, with a rainfall between 5 and 10 cm (http://www.ncdc.noaa.gov/climate-information).

We collected bat guano from the attic of a storage barn belonging to Oregon State University in Corvallis, OR. The guano was tested for Pd prior to installation of the experiment and autoclaved as a precautionary measure to prevent potential harboring of active Pd and other pathogens. We measured 1 gram (SD 0.018 g) of guano and placed each sample in individual petri dishes. One milliliter of genomic DNA (gDNA) (ATCC MYA-4855D) was spiked onto the guano in 10-fold serial dilutions ranging from 1 million fg to 1,000 fg, including a negative control. After pouring the inoculum, samples were swirled to spread the inoculum evenly on the guano pellets. Each concentration was prepared in triplicate for a total of 15 samples per site. We placed the 15 petri dishes in identical trays at each site, started our sampling on day zero (deployment day) and continued sampling every 14 days, ending on day 70. Trays were placed on the ground, outside of buildings

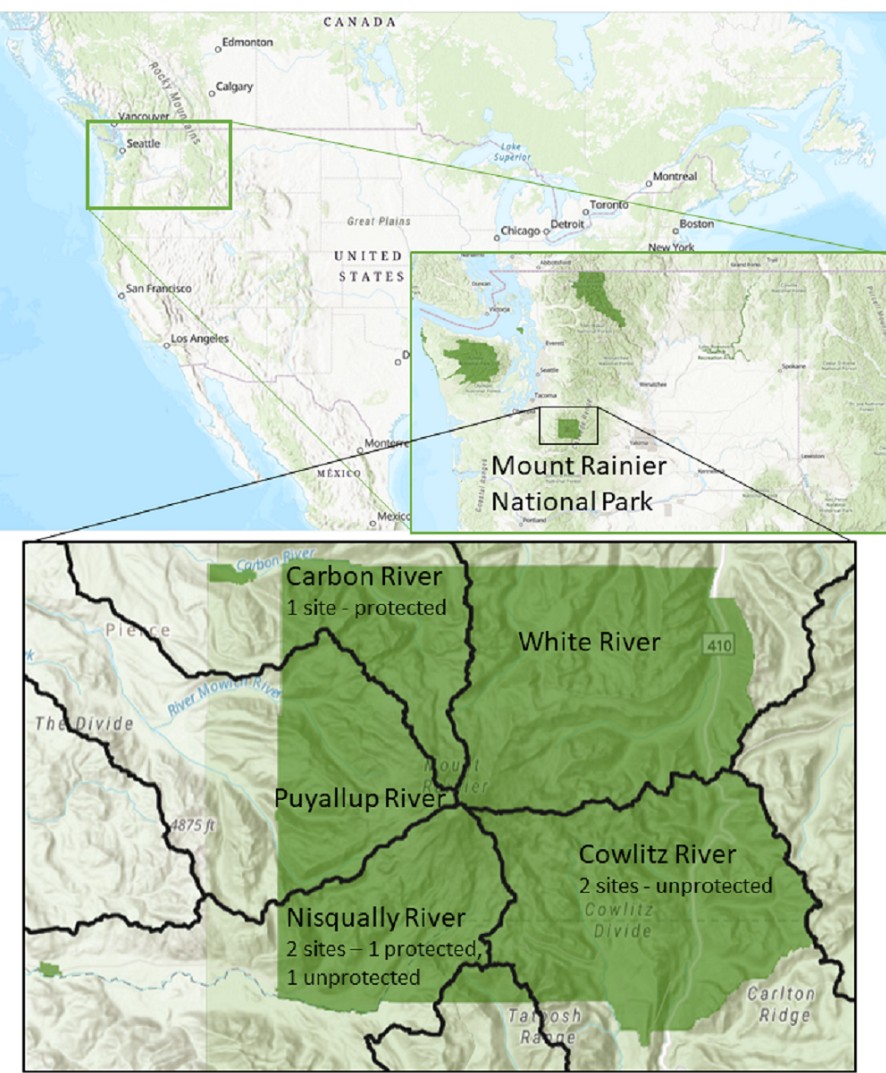

**Figure 1** **Locations of experiment deployment at five occupied bat colonies in three watersheds in Mount Rainier National Park, Washington State, USA.** Identified by 10th field Hydrological Unit Code (HUC): Carbon River HUC 1711001401, Cowlitz River HUC 1708000401, Nisqually River HUC 1711001501. Data Sources: US Geological Survey, National Park Service, ESRI.

and structures where bats have been reported but adjacent to locations where bats exited to ensure cross-contamination did not occur. Each tray was also labeled with a sign visible from 3 m indicating ''Research in Progress—Do Not Disturb'' to ensure the experiment was not disturbed by visitors or workers. We thus collected 15 samples per each of five sites during six different sampling events from July 10th 2018 and September 17th 2018 for a total of 450 fecal samples (mean = 0.13 ± 0.06 g/sample). At each site, we wore a new pair of gloves for each concentration to collect the pellets to avoid cross contamination among different concentrations. Collected samples were placed in 2.0 mL Eppendorf safe-lock tubes and stored on ice. Immediately after each sampling event, samples were

shipped overnight to Oregon State University where they were stored at $-20\,°C$ until DNA extraction.

## Quantifying gDNA

DNA was extracted from fecal pellets using the DNeasy PowerLyzer Powersoil Kit following the manufacturer's instructions. One extraction blank was included with each batch every 25 samples to be used as non-template controls. Mass of all guano samples was measured prior to extraction with an average of $0.13 \pm 0.06$ g. DNA was amplified in a probe-based assay targeting the intergenic spacer (IGS) region of *Geomyces destructans* (Muller et al., 2013). Reactions consisted of 12.5 ul TaqMan Environmental MasterMix 2.0 (Life Technologies, Carlsbad, CA), forward primer nu-IGS-0169-5'Gd and reverse primer nu-IGS-0235-3'Gd at a final concentration of 400 nM, TaqMan FAM-labeled probe at a final concentration of 200 nM and 5 µl of DNA template. Using Pd gDNA, we prepared a 4-point standard curve, diluted 10-fold, with the highest concentration containing 1,000 fg gDNA and the lowest point containing 1 fg DNA. The quantification of DNA was performed using an ABI PRISM 7500 Fast real-time PCR system (Applied Biosystems, Foster City, CA) with the following cycling conditions: initial activation $95\,°C$ for 10 min; denaturation $95\,°C$ for 15 s and annealing, and extension $60\,°C$ for 60 s, with a total of 40 cycles.

All standards were run in triplicate to generate reference curves to control for consistency across plates and check for variation that can be caused by low amounts of template (*Verant et al., 2016*). Amplification efficiency of qPCR reactions calculated as an average across all plates was 90%. This lower percentage is due to high variation coming from our lowest standard 1 fg. We analyzed all samples collected in triplicate, and they were reported as positive when 2 out of the 3 wells amplified within 40 cycles. Our limit of quantification corresponded to 37 Ct value of our lowest standard (1 fg). However, our limit of detection was up to 39 Ct as we detected amplification for the target Pd sequence at this higher Ct value. No amplification was seen in the extraction blanks.

## Data analysis

We evaluated both the rate of DNA degradation and the change in the probability of detection through time. To evaluate the rate of DNA degradation, we used log-transformed DNA quantity as measured with qPCR as the dependent variable in a linear model with normal error distribution. To facilitate log-transformation, we added 0.195 fg to all DNA concentrations, which represents the lower limit of detection using qPCR. To evaluate the change in the probability of detection, we used logistic regression with Pd positive/negative status as the dependent variable. In both models, our predictors included the initial quantity of the inoculum (categorical with four levels), field site where inoculated guano was placed (categorical with five levels), days since inoculation (continuous), mass of guano sampled (continuous), and an interaction term for days since inoculation and site to account for variable rates of DNA degradation by site. Based on our model for DNA quantity through time, we projected the expected measured Pd quantity for 100 days since deposition, across the four initial concentrations and five sampling locations, each with its own degradation rate to characterize the variable rate of degradation among sites. All statistical analyses
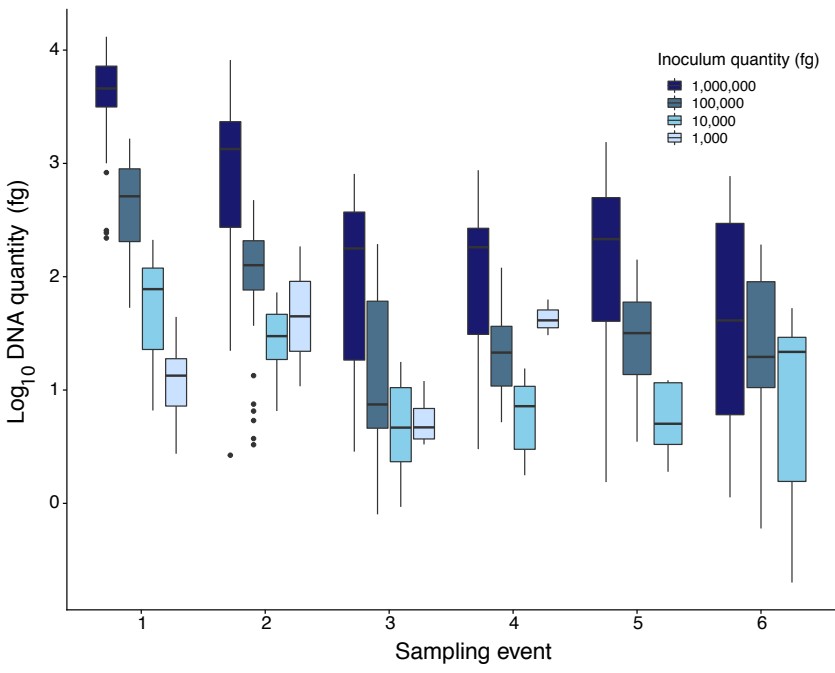

**Figure 2** **DNA quantity of *Pseudogymnoascus destructans* for all experimental sites sampled during six different times.** Highest inoculation quantity (1 million fg) is represented by dark blue bars. Lowest inoculation quantity (1,000 fg) is represented by the light blue bars.

were performed in R (Version 3.4.1, 2017-06-30) using packages plyr, magrittr, ggplot2 and cowplot (*Wickham, 2011*; *Wickham, 2016*; *Milton & Wickham, 2014*; *Wilke, 2019*)

## RESULTS

Quantity of Pd DNA in guano decreased through time with different decay rates among sites (Fig. 2). The rate of degradation varied from 1.6% (Nisqually Site 1) up to 8.2% (Ohanapecosh Site 2). Degradation rates at the protected sites (i.e., Nisqually Site 1 1.6% and Carbon Site 2.9%) were lower than those in unprotected sites (i.e., Nisqually Site 2 and Ohanapecosh Site 1 and 2), which presented a higher degradation rate through time (Rate$_{Nisqually\ Site\ 2}$ = 6.3%; Rate$_{Ohana\ Site\ 1}$ = 7.7%; Rate$_{Ohana\ Site\ 2}$ = 8.2%) (Fig. 2).

Samples inoculated with the two highest quantities (1 million fg and 100,000 fg) amplified during all sampling events at all sites (Fig. 3 and Table 1). After 70 days, 10 and 8 out of 15 samples (66 and 53%) amplified (Table 1). During the first sampling event, we detected Pd DNA from 9 out of 15 samples (60%) inoculated with the lowest quantity (1,000 fg) amplified in 4 out of 5 sites. After 13 days, the probability of detection in the lowest initial quantity decreased to 6%, with amplification from 1 of 15 samples (Fig. 4). However, we observed variation in detection probability based on the site. We detected Pd in samples inoculated with our lowest quantity (1,000 fg) at Nisqually Site 2 for 30 days, and the Carbon Site 1 for 42 days. We found evidence that the probability of detection is explained by the interaction between sampling time and site for 3 out of
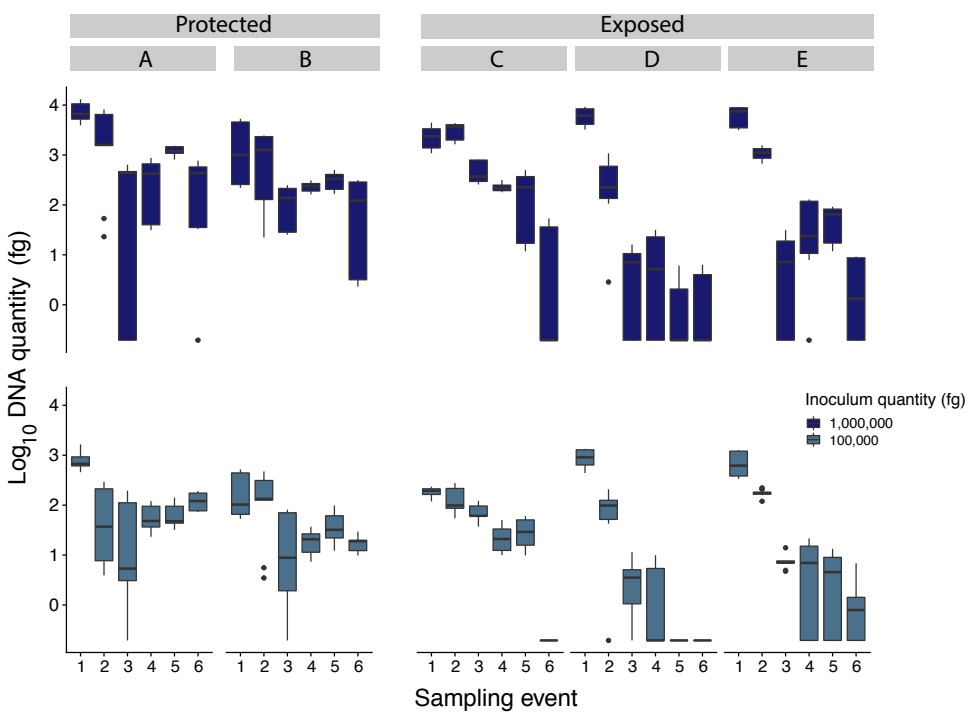

**Figure 3** Comparison of *Pseudogymnoascus destructans* DNA quantities recovered from samples inoculated with 1 million and 100,000 fg among all sampling sites (indicated by caps letters). Sites A and B were protected sites, with less exposure to wind and solar radiation while sites C, D, and E, were more exposed to wind and solar radiation.

**Table 1 Number of samples (percentages) that amplified during each sampling event. Total number of samples per sampling event = 15.**

| Sampling Inoculum quantity (fg) | 1 Day 1 | 2 Day 13 | 3 Day 29 | 4 Day 42 | 5 Day 56 | 6 Day 70 |
|---|---|---|---|---|---|---|
| 1,000,000 | 15 (100%) | 15 (100%) | 12 (80%) | 14 (93%) | 14 (93%) | 10 (66%) |
| 100,000 | 15 (100%) | 15 (100%) | 15 (100%) | 12 (80%) | 11 (73%) | 8 (53%) |
| 10,000 | 15 (100%) | 13 (86%) | 8 (53%) | 6 (40%) | 3 (20%) | 5 (33%) |
| 1,000 | 9 (60%) | 1 (6.6%) | 2 (13.33%) | 1 (6.6%) | 0 | 0 |

the 5 sites ($\beta_{\text{Nisqually Site 2}} = -0.108$; $P => 0.0001$, $\beta_{\text{Ohana Site 1}} = -0.09$; $P => 0.0001$, $\beta_{\text{Ohana Site 2}} = -0.011$; $P => 0.0001$) (Table 2). Mass had a significant effect on the amount of DNA detected ($\beta = 6.93$; $P => 0.0001$) and in the positive detection of samples ($\beta = 7.58$; $P = 0.0001$). All controls were negative for Pd amplification. Projections based on the models suggested that although DNA degradation rates vary among sites, it is possible to detect extracellular Pd after 100 days for samples initially inoculated with the highest quantities (1 million and 100,000 fg) or approximately 80 days for the lowest quantities (10,000 and 1,000 fg) of Pd DNA (Fig. 5).
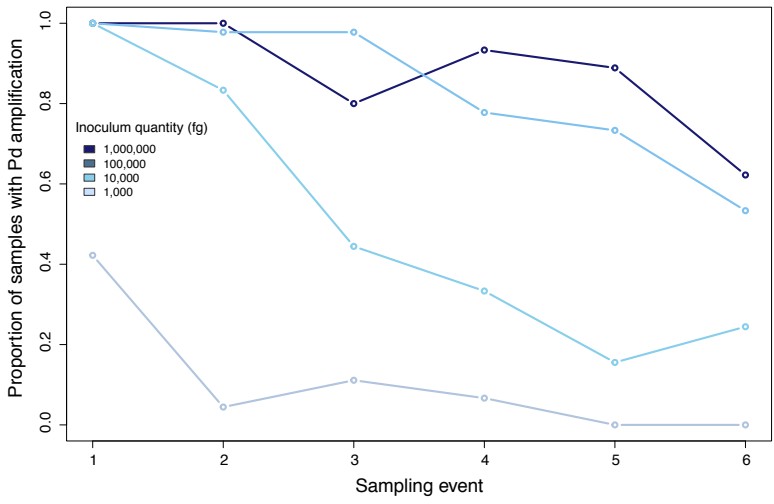

**Figure 4** Proportion of positive samples of *Pseudogymnoascus destructans* detected during all sampling events.

**Table 2   Amount of DNA (fg) that can be detected at time 0 per site sampled with a mean weight of sample.**

| Inoculum quantity (fg) | Nisqually Site 1 | Nisqually Site 2 | Ohanapecosh Site 1 | Ohanapecosh Site 2 | Carbon River Site 1 |
|---|---|---|---|---|---|
| 1,000,000 | 421.48 | 1708.71 | 570.04 | 1380.02 | 441.51 |
| 100,000 | 73.25 | 296.98 | 99.07 | 239.85 | 76.73 |
| 10,000 | 7.18 | 29.11 | 9.71 | 23.51 | 7.52 |
| 1,000 | 1.15 | 4.66 | 1.55 | 3.77 | 1.20 |

## DISCUSSION

Pd is a continued threat to bat populations and diversity as it spreads across North America (*Alves, Terribile & Brito, 2014*). The most recent report of Pd and WNS in the western United States from July 2019 suggests Pd is spreading and common species unique to western North America are vulnerable to WNS, e.g., western long-eared myotis. Testing for WNS can be done with invasive methods such as capturing and swabbing individual bats and collecting tissue biopsies from the wings to confirm infection, but this can be expensive, time consuming, requires specialized animal handling training and vaccinations, and imposes stress to the animal and a risk of unhealed lesions. If sufficiently effective, noninvasive monitoring of Pd DNA using guano could save time and costs to allow for larger surveillance efforts for early Pd detection while avoiding handling of bats. While noninvasive monitoring does not confirm disease status, it can be a useful tool to mobilize early detection and rapid response strategies. DNA in guano degrades through time in a manner affected by exposure to the ambient environment that varies among sites (*Verant et al., 2018*). Here our goal was to establish baseline expectations of degradation rates and detectability of extracellular Pd DNA in guano left under ambient environmental conditions near the colonizing front of Pd in western North America.

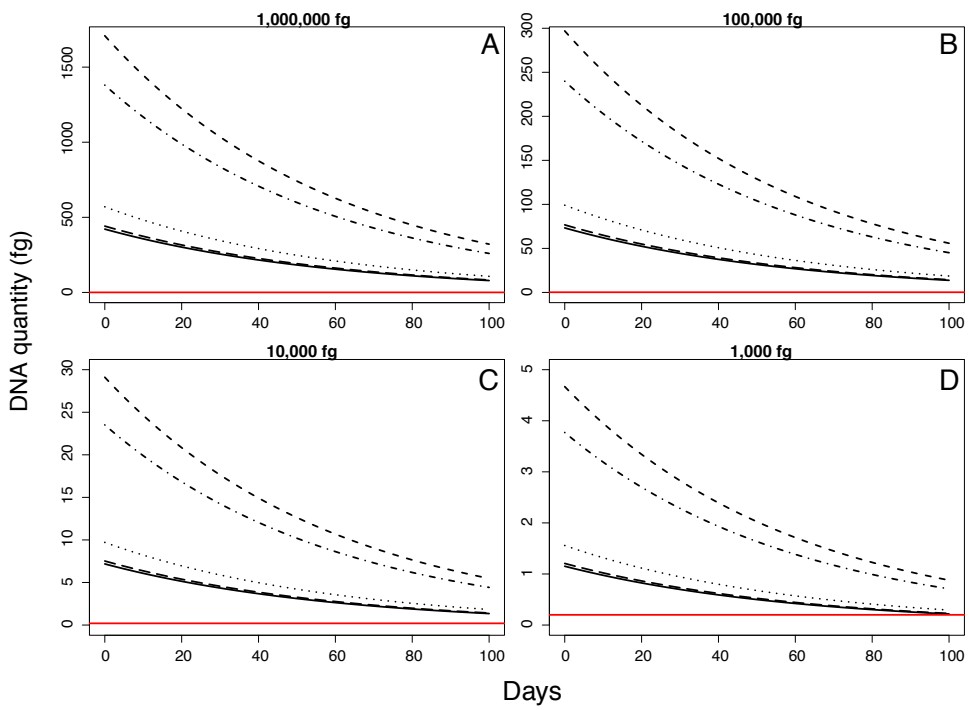

**Figure 5   Model projections for detectable quantity of *Pseudogymnoascus destructans* for 100 days according to different quantities of inoculum.**

Quality of DNA that can be recovered from bat guano samples varies through time. Samples collected more than 10 years prior to the time of analysis delivered less DNA (*Zinck, Duffield & Ormsbee, 2004*) than samples that have been just recently collected (*Vege & McCracken, 2001*). In our study, a higher proportion of samples collected during the first sampling events amplified, after which amplification declined until the end of the experiment. Degradation of extracellular DNA depends on environmental conditions such as the level of water, oxygen and temperature (*Willerslev, Hansen & Poinar, 2004*). In this study, the initial quantity of inoculum and mass were variables related to the probability of DNA detection through time. Samples inoculated with the highest quantities successfully amplified from all sampling events while samples initially inoculated with lowest quantities (10,000 and 1,000 fg) amplified in lower proportions. If a sample of ∼0.14 g is sampled from ∼1 g of guano inoculated with the lowest quantity used in this experiment (1,000 fg) the total amount that can be detected will be ∼140 fg. However, using our results we estimated that detection can be as low as 1.15 fg for one of the sites. The amount of DNA that can be detected changed among sites and sampling events.

We found that extracellular Pd DNA in summer can be detected for up to 42 days when inoculated at 1,000 fg, and for up to 70 days after deployment with estimates indicating DNA can be detected for approximately 100 days if initial deposition quantities are among our higher inoculations (1 million and 100,000 fg). In our study, DNA degraded faster in sites exposed to sunlight without overhead cover. Exposure to direct solar radiation

can result in DNA degradation by denaturation in exposed samples (*Schuch et al., 2017*). Similarly, high temperatures generated by direct radiation can drive a high degradation of DNA in contrast to low temperatures (*Nagler et al., 2018*). In contrast to cold environments where nucleic acids can be stored for long-term due to the decrease of reaction rates by an order of magnitude for every degree drop in temperature (*Smith et al., 2001*).

The minimum amount of Pd DNA detected in guano in our study was 0.195 fg, similar to the amount reported by *Verant et al. (2018)* (0.28 fg) from a study done in hibernacula sediments. The amounts of Pd reported from guano and sediments were lower than loads reported from bat swabs and tissue samples. During hibernation *Langwig et al. (2015)* reported loads between 1,000 and 10 fg and loads during swarming and maternity as low as 1 fg. However, different species can show different maximum fungal loads through time (*Langwig et al., 2016*; *Frick et al., 2017*). Pd in guano is hypothesized to originate from ingestion of the fungus by bats during grooming, as bats with the disease groom more often than uninfected bats. As Pd seems to resist gastrointestinal passing, movement of bats between hibernacula can lead to disperse of Pd spores to new locations (*Brownlee-Bouboulis & Reeder, 2013*; *Ballmann et al., 2017*). As degradation of DNA can be affected by environmental variables that change per location, we recommend use of an integrated approach that includes monitoring of multiple sites and their environmental variables to inform agencies and managers about the role those variables are playing on the rates of DNA degradation, and how it affects monitoring and surveillance studies. Properly designed sampling of hosts and their pathogens will improve our chances for an early detection of Pd in bats and potential emergence in new areas. Wildlife disease surveillance must provide information for agencies in a timely manner to be able to generate appropriate management actions especially in the case of novel pathogens (*Sleeman et al., 2019*).

## CONCLUSIONS

Throughout this study, we provided experimental evidence that validate the detection of Pd in guano as an effective non-invasive technique for pathogen surveillance. The probability of Pd detection in guano is affected by environmental factors that are site specific as well as initial quantity of inoculum and sampled mass. There is still a need to determine the source of Pd DNA detected in guano as it could be extracellular DNA fragments, intracellular dead Pd cells, or intracellular viable Pd cells, that can affect the time of degradation and the probability of detection. If positive results of Pd are detected, we recommend including and notifying the records immediately at http://www.whitenosesyndrome.org as well as to the Fisheries and Wildlife unit working in your locality.

## ACKNOWLEDGEMENTS

We owe thanks to the bat monitoring team at Mount Rainier National Park (Alaiya Cave, Dacey Clark, Alexis Levorse, Colin Woodbury) for their effort sampling bats in the park, for their input, and for the provisioning of samples. Thank you to the Levi and Garcia lab at Oregon State University for the use of their lab equipment.

### Funding

Funding was provided by the National Park Service White-nose Syndrome Service wide Fund Source and Greening Youth Foundation Historically Black Colleges and Universities Internship to Alaiya Cave. The funders had no role in study design, data collection and analysis, decision to publish, or preparation of the manuscript.

### Grant Disclosures

The following grant information was disclosed by the authors:
National Park Service White-nose Syndrome Service wide Fund Source.
Greening Youth Foundation Historically Black Colleges and Universities Internship to Alaiya Cave.

### Competing Interests

The authors declare there are no competing interests.

### Author Contributions

- Jenny Urbina conceived and designed the experiments, performed the experiments, analyzed the data, contributed reagents/materials/analysis tools, prepared figures and/or tables, authored or reviewed drafts of the paper, approved the final draft.
- Tara Chestnut and Jenn Allen conceived and designed the experiments, performed the experiments, contributed reagents/materials/analysis tools, authored or reviewed drafts of the paper, approved the final draft.
- Donelle Schwalm conceived and designed the experiments, contributed reagents/-materials/analysis tools, authored or reviewed drafts of the paper, approved the final draft.
- Taal Levi conceived and designed the experiments, analyzed the data, contributed reagents/materials/analysis tools, prepared figures and/or tables, authored or reviewed drafts of the paper, approved the final draft.

### Data Availability

Dataset: https://ir.library.oregonstate.edu/concern/datasets/v405sh53x.

### Supplemental Information

Supplemental information for this article can be found online at http://dx.doi.org/10.7717/peerj.8141#supplemental-information.

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
