# Peer review of "Experimental evaluation of genomic DNA degradation rates for the pathogen Pseudogymnoascus destructans (Pd) in bat guano"

_PeerJ, doi:10.7717/peerj.8141_

## Round 0.1 · original submission · Major Revisions

I invite you to address the comments from two reviewers. Thank you.

Reviewer 1 ·

Basic reporting

The manuscript entitled "Experimental evaluation of genomic DNA degradation rates for the pathogen Pseudogymnoascus destructs (Pd) in bat guano" describes an experiment aimed at measuring Pd in bat guano over time.

The main laboratory procedure in this manuscript, qPCR assay for Pd, is not properly described and/or cited. There are several published qPCR assays for Pd in the literature, it seems that the authors likely used the method described in Muller et al. (2013) -- however, it is not cited. Additionally the probe used for qPCR is not described either.

If the widely used Muller et al (2013) assay was used herein, then there are some problems with limit of detection described in this paper, as the lowest limit of detection in the Muller et al (2013) qPCR assay was reported to be 3.3 fg. Here the authors claim they could measure 0.195 fg, which is outside the detection limits of the assay. Moreover, the authors state that a standard curve was run for each qPCR experiment using purified gDNA from Pd in 1:10 serial dilutions from 1000 fg/ul to 1 fg/ul -- and that 5 ul was used for each qPCR assay --> so that means the limit of detection in this assay cannot be below 5 fg, the lowest concentration in the standard curve.

Lines 137-180: What is a positive result? what is a negative result?

Experimental design

The research question addressed here was to measure the genomic DNA degradation rate in bat guano over time. The authors spiked Pd DNA into autoclaved bat guano and then incubated these samples in ambient temperature at multiple roosting sites. DNA was measure every 14 days from guano samples in triplicate. While the main research question is to answer DNA stability/detection in guano samples, there are no laboratory controls to give the results context, ie is DNA stability in bat guano better or worse than leaving a sample on a bench top for 70 days? different temperatures, etc.

There are several occasions where the authors seem to use quantity (fg) interchangeably with concentration (fg/ul) -- these are important details and do not mean the same thing. For example, the methods say that 5 concentrations of DNA (fg/ul) were inoculated into guano? What was the volume? How did you inoculate? What was total fg of DNA per mg of guano? How does the DNA extraction yield impact the results? These details are necessary in order to quantify how much DNA can be extracted from bat guano in the first place, e.g. taking into account yield loss during DNA extraction might influence how you interpret the results.

Validity of the findings

Underlying qPCR data not provided. At a minimum, the standard curve used for interpretation of the qPCR results should be provided.

Figures 1 and 2 don't really add anything to manuscript.

Reviewer 2 ·

Basic reporting

L227. Citation needed

Experimental design

L101-109. It is odd that actual environmental conditions were not measured at the five sampling locations, particularly temperature and relative humidity. It is also unclear how often guano samples from typical sampling sites are exposed to direct solar radiation or rain. Many roost sites (and consequently the guano) are relatively protected from each potential source of DNA degradation.

Validity of the findings

To my knowledge, it is unclear from the current literature whether Pd is viable from guano. If it is indeed viable, then one would posit that the DNA would persist longer.

Additional comments

This manuscript addresses an important question of the ability to detect Pd over time from environmental samples such as guano.

Minor Comments/Edits
L30. unclear who is doing the positing. Perhaps change to "it has been posited"?
L42. Change smallest to lowest
L53. Change cryophilic to psychrophilic
L66. Change bat to bats
L75. Citations confusing. As written it appears that these citations refer to Pd studies when in fact they are from other pathogens. Perhaps put these citations earlier in sentence (e.g. after the word guano) or rewrite for clarity.
L76. Muller et al. 2013 is the best citation for the qPCR. Should also be cited in the methods section. Muller, L. K., J. M. Lorch, D. L. Lindner, M. O’Connor, A. Gargas, and D. S. Blehert. 2013. Bat white-nose syndrome: a real-time TaqMan polymerase chain reaction test targeting the intergenic spacer region of Geomyces destructans. Mycologia 105:253-259.

L83. Unclear where to put this but sampling from guano under real conditions may also detect Pd from the environment as it is shed off of bats.

An alternative to sampling guano could be to sample cave/hibernacula/bat house substrates, with Langwig et al. 2015 demonstrating the ability to detect Pd from cave walls and Dobony & Johnson 2018 from the surfaces of bat boxes.

Langwig, K. E., J. R. Hoyt, K. L. Parise, J. Kath, D. Kirk, W. F. Frick, J. T. Foster, and A. M. Kilpatrick. 2015. Invasion dynamics of white-nose syndrome fungus, Midwestern United States, 2012-2014. Emerging Infectious Diseases 21:1023-1026.

Dobony, C. A., and J. B. Johnson. 2018. Observed resiliency of little brown myotis to long-term white-nose syndrome exposure. Journal of Fish and Wildlife Management 9:168-179.

L113. More details are needed about how the inoculation was actually done. With a pipette? The DNA was dripped onto the pellet? Spread evenly? What was the process?

L116. "on a the"?
L118. Change ending to ended
L123. Change inside to in
L131, 132. Change Taqman to TaqMan
L133. Are these the conditions of Muller et al. 2013? If so, cite here.
L137. What was your cycle threshold value? any amplification before Ct of 40?
L177. Not sure what you mean by "weight", guano sample weight? Also, I believe it should be referred to as mass?
L190. This is misleading as stated. Collecting wing tissue is only needed for confirmation of WNS, not detection of Pd. Noninvasive sampling such as guano testing cannot confirm WNS. Thus, a wing punch is not typically used for Pd detection.
L204. Again, I'm confused about what weight means here
L205. The point of this sentence is unclear. And isn't intracellular DNA only repaired in living cells?
L207. Delete were
L207. Change during to from
L213. A more complete description of the sites would be helpful. It was surprising to see that some of these sites were in the sun.
L220. Change is to was
L233-234. This is only true if the WNS community can agree on testing standards for Pd assays, Ct values, etc. Currently some states are trying to refute Pd detections for a variety of reasons. For example, if guano sampling typically only detects low levels of Pd then some states may not consider these to be true detections.

Figure 2. Was this modeled data? If so, you should state that.
Table 1. Unclear what the values for each site are referring to? The legend needs more details.

---

## Round 0.2 · accepted · Accept

Congratulations. I look forward to your future work on the white-nose syndrome in bat.

Reviewer 1 ·

Basic reporting

no comment

Experimental design

no comment

Validity of the findings

no comment

Additional comments

The authors have addressed my major concerns with this revision.

Reviewer 2 ·

Basic reporting

no comment

Experimental design

no comment

Validity of the findings

no comment

Additional comments

The authors have addressed all of my concerns.